

# Evaluating the Performance of Objective Functions and Regional Climate Models for Hydrologic Climate Change Impact Studies: A Case Study in the Eastern Mediterranean

Ioannis Sofokleous[1], George Zittis[2], Gerald Dörflinger[3], and Adriana Bruggeman[1]

[1] Energy, Environment & Water Research Center (EEWRC), The Cyprus Institute, Nicosia, Cyprus.

[2] Climate and Atmosphere Research Centre (CARE-C), The Cyprus Institute, Nicosia, Cyprus.

[3] Water Development Department, Nicosia, Cyprus.

*Correspondence to*: Ioannis Sofokleous (i.sofokleous@cyi.ac.cy)

**Abstract.**

The robustness of hydrological models used in projections of future fresh water resources is compromised due to non-stationary climate conditions. This study aims to (i) develop a method for selecting a skillful hydrological model parameterization under changing climate conditions and (ii) apply a calibrated hydrological model to assess

streamflow projections for 38 mountain watersheds in the eastern Mediterranean island of Cyprus over the next decades (2030-2060). A matrix-based approach was developed to evaluate six objective functions by eight performance measures. Using the GR4J hydrological model, evaluation matrices were computed for multiple 5-year simulation runs covering 1980–2015. The matrices covered 14 model calibrations and 182 validations in total, as well as 4 sets of validations under different climate change conditions, for each watershed. Based on the matrix method,

the Nash-Sutcliffe Efficiency with square-root transformed streamflow resulted in the best performance for streamflow simulations in Mediterranean watersheds experiencing drying trends. This method is transferable and can be applied in different climate regions to identify the most suitable objective function and model parameterization for hydrologic climate impact assessments. Eighteen Regional Climate Models (RCMs) were bias-corrected, downscaled to 1 km and used to simulate streamflow with GR4J for 1980-2010. Nine RCMs underestimated the fraction of wet

period precipitation (60-73% instead of 82% of annual precipitation), causing streamflow biases up to 40%. The remaining nine RCMs selected for the study simulated the seasonal precipitation cycle accurately. The median of future projections showed a 6% reduction in precipitation and a 17% reduction in streamflow. In the worst case, reductions could reach 16% and 39%, respectively. Notably, during the driest years, streamflow reductions could reach 70% relative to the driest years in the past. Our findings suggest that terrestrial water resources in the eastern

Mediterranean may significantly deteriorate in the coming decades.





## 1 Introduction

Historical records show that renewable fresh water resources in the Mediterranean region are decreasing (Gudmundsson et al., 2021). The second half of the 20[th] century has been characterized by decreasing trends in rainfall
and river flow and increasing frequency of droughts across catchments in the Mediterranean (Vicente-Serrano et al., 2014; Marques da Silva et al., 2015; Myronides et al., 2018; Masseroni et al., 2021). The increasing frequency and intensity of water scarcity episodes in the region require the design of adaptation plans, which must take into account climate-driven projections of future water resources (Tramblay et al., 2020). Conceptual hydrological models are most often used for climate change impact studies, however, their capacity to simulate streamflow under changing climate
conditions is being questioned (Refsgaard et al., 2014; Kang Ji et al., 2023).

For the calibration of hydrological models to be used in impact assessments, the aspect of transferability for future climate conditions of model parameters that were calibrated for past conditions is under discussion. The Differential Split-Sample Test (DSST), described by Klemes (1986), is commonly used to test how well a hydrological model performs for different climate conditions than the conditions for which the model was calibrated (Refsgaard et al.,
2014). This approach is used considering changes in temperature, precipitation or both (e.g., Vaze et al., 2010; Thirel et al., 2015; Dakhlaoui et al., 2017). However, a number of studies in different climate zones found that model parameter transferability under DSST is less promising for transition towards drier conditions than for transition towards wetter conditions (e.g., Broderick et al., 2016; Le Coz et al., 2016; Yang et al., 2020). The model robustness for the transition to drier conditions has also been found to be lower in catchments with high runoff skewness and
aridity, commonly found in the Mediterranean and dry climates (Munoz – Castro et al., 2023; Guo et al., 2020). Based on this finding, climate change impact studies recommend calibrating models using sub-periods that have similar annual mean precipitation and temperature to the future periods, rather than calibrating over the entire past period (Dakhlaoui et al., 2019). Alternatively, Kang Ji et al. (2023) recommended considering future simulations credible only when the changes in mean temperature and precipitation in future climate are less than certain values, e.g.,
precipitation change less than 10% and temperature change less than 1.75°C.

The selection of the evaluation measure to be used as an objective function for calibrating a hydrological model could affect the assessment of climate impact. For instance, Fowler et al. (2018) found that models calibrated on the Refined Index of Agreement (Wilmott et al., 2011), which uses the sum of absolute errors, and Kling Gupta Efficiency (KGE; Kling et al., 2012) computed individually per year, performed better than models calibrated on squared-error based
measures. The same study suggests that streamflow data for the computation of NSE (Nash-Sutcliffe Efficiency; Nash & Sutcliffe, 1970) or KGE values should be transformed when simulations are intended for drier future conditions. The square root transformation of streamflow, compared to the inverse transformation, was found to be the most suitable approach for a balanced optimization of both high and low flows, with a small loss of confidence for extremely high or low flow conditions (Seiller et al., 2017). Munoz-Castro et al. (2023) found that the choice of objective function
had a larger effect on model performance for high-aridity and low runoff coefficients than on model performance for wetter climates, from a comparison of 12 functions.



The impact of climate change on terrestrial water resources has been studied on global and regional levels using hydrological models forced by different global and regional climate models (GCM/RCM). In a global analysis, Hageman et al. (2013) showed that southern Europe and the Middle East will be most severely impacted by future

runoff reductions. Roudier et al. (2016) found that model projections of an increase in drought magnitude and duration were more robust, i.e., with lower model spread, for southern Europe than for the rest of the continent for a global warming level of 2°C since pre-industrial. Similarly, Marx et al. (2018) found that, in a 3°C global warming scenario, low flows, defined by the flow threshold exceeded 90% of the time, in the Mediterranean region will become even lower, with up to -35% reduction.

These changes in the terrestrial water resources are driven by changes in temperature and precipitation. For the Mediterranean, Zittis et al. (2022) reported that the current regional warming rate is 0.45°C per decade, nearly two times higher than the global average trend (0.27°C per decade). Cos et al. (2022) found that simulations from the Coupled Model Intercomparison Project (CMIP, at 1° spatial resolution) project a stronger warming in the Mediterranean, relative to the global mean change, particularly in the summer, which could range from 1.8°C to an

alarming 8.5°C by the end of the century. The range of these projections corresponds to the Radiative Concentration Pathways (RCP) examined at different levels, e.g., RCP2.6, RCP4.5 and RCP8.5. After analyzing a large ensemble of high-resolution (0.11° ~ 12km) RCM simulations for the Mediterranean, Zittis et al. (2021a) highlighted an annual precipitation reduction of up to 10% for the first half of the 21st century and reductions up to 20%-40% for the second half, particularly for the southern and eastern areas. In addition to the reduction in average annual amounts, an increase

in the number of consecutive days without rain in the region is expected (Reymond et al., 2019). Two major drivers for the drying trend in the Mediterranean are a robust change in the upper-tropospheric large-scale circulation and the reduction of the gradient in land-sea temperatures (Tuel & Eltahir, 2020). Due to the reduction in precipitation and streamflow volumes, the aquatic state of streams could be altered, affecting aquatic habitats and riparian ecosystems (Gallart et al., 2012; Martínez-Fernández et al., 2018).

Assessing the impact of climate change on fresh water resources through hydrological modelling requires knowledge of the performance limits of these models in the context of a changing climate. This is especially important for the drier environments of the Mediterranean region, where the observed and projected warming rates are above the global averages and the precipitation projection trends are mostly negative. This study aims to examine the role of objective functions and climate change signals on the capability of a conceptual hydrological model to quantify the impact of

climate change on future streamflow. The specific objectives are: (i) to develop a method for evaluating the performance of different objective functions for hydrological model calibration under a changing climate, (ii) to select the most skillful model parameterization for applications under future climate conditions, (iii) to bias-correct, downscale and evaluate the performance of a large RCMs ensemble for streamflow simulations, and (iv) to apply the optimized hydrological model with selected RCMs for 38 mountain watersheds in Cyprus, and assess the mid-term

future (2030-2060) impact on the island's water resources.



## 2 Data and Methods

### 2.1 Model calibration for changing climate conditions

#### 2.1.1 Evaluation of objective functions

Six objective functions were used independently for six separate hydrological model calibrations. These functions are based on the NSE and KGE criteria, each computed with three types of transformation of the streamflow values: 1) no transformation, 2) square root and 3) natural logarithm (NSE, NSEsqrt, NSElog, KGE, KGEsqrt and KGElog). To understand how the calibrated models, optimized on these objective functions, reproduced the different attributes of the observed streamflow, the six functions were also used as evaluation measures. The percent bias (PBIAS) of total

streamflow was used as a seventh evaluation measure. Finally, a Composite Scaled Score (CSS; eq. 1), which is used for a relative performance comparison of the objective functions, is computed using all seven evaluation measures. This results in a matrix of six objective functions and eight evaluation measures, for a simultaneous evaluation of the objection functions per watershed. To evaluate the overall performance of the objective functions for all watersheds, a summarizing 6x8 matrix, comprised of the median values of the 38 watersheds for each measure and each objective

function was used.

The CSS is computed by averaging the normalized values of the seven evaluation measures as follows (Sofokleous et al., 2021):

$$CSS_i = \frac{1}{N_s}\sum_{s=1}^{N_s}\left(\frac{Y_{s,i}-Y_{s,worst}}{Y_{s,best}-Y_{s,worst}}\right) \tag{1}$$

where $i$ is the index identifying the objective function, $s$ is the index of the evaluation measure out of a number of $N_s$

(seven) measures, $Y_{s,i}$ is the value of measure $s$ obtained by objective function $i$ and $Y_{s,worst}$ and $Y_{s,best}$ are the worst and the best values for measure $s$ obtained by the six objective functions. The score ranges between 0 and 1, with 1 corresponding to the best performance. The CSS has been successfully applied in performance inter-comparisons of different model configurations and methods (Sofokleous et al., 2024; Citrini et al., 2024).

#### 2.1.2 Selection of objective function under changing climate conditions

Calibration was performed for all moving 5-year windows, within the calibration period, applying all six objective functions. The hydrologic year preceding each 5-year window was used as a warm-up year. Validation was conducted for each calibration window and each corresponding calibrated model based on the six objective functions across 5-year windows within an independent validation period. The objective functions were evaluated for, firstly, all 5-year

calibration and 5-year validations pairs, and secondly, pairs of 5-year calibration and 5-year validation periods corresponding to different changes in climate conditions. The effect of climate conditions change from the calibration to the validation periods was based on the relative change of total precipitation and average temperature from the calibration to the validation. The temperature minimum increase to select a set of calibration and validation run was 0.7 °C. This temperature threshold was based on the observed past changes in the study area and the expectation of

warmer future conditions (validation) relative to the past climate (calibration). The calibration-validation pairs of runs



satisfying this temperature criterion were grouped in four precipitation change ($\Delta P$) classes, representing conditions of increased wetness and increased dryness at different thresholds from the calibration to the validation period, i.e., $\Delta P > 15\%$, $5\% < \Delta P \leq 15\%$, $-5\% < \Delta P \leq 5\%$, and $\Delta P \leq -5\%$.

The matrix-based evaluation of the six objective functions was performed for all 5-year calibration and all 5-year validation periods and for the 5-year validation periods matching each of the four precipitation change classes. The objective function that achieved the highest score (CSS), among the six functions, on average for both the calibration and the validation periods, was selected and used for the model optimization for future streamflow simulations.

### 2.1.3 Selection of model parameterization for future simulations

For the selection of the optimal model parameterization, model runs obtained using the selected objective function were evaluated. For each watershed, the parameter set that achieved the highest value of the selected objective function during the validation period, corresponding to the projected precipitation change class ($\Delta P$) for the study area, was used in the hydrological model for future simulations. Unlike the standard split-sample calibration–validation methodology, where the parameter set is selected based solely on calibration performance, this approach selects the parameterization based on model performance in both the calibration and validation periods, and in relation to observed changes in temperature and precipitation between the two periods.

### 2.2 Study area and observational data

The streamflow simulations for the past and the future are conducted in 38 watersheds in Cyprus. Cyprus is an island in the eastern part of the Mediterranean Sea, at latitude 35°N and longitude 33°E. The climate of the island is characterized by the typical annual and interannual variability in precipitation of Mediterranean climates (Hoerling et al., 2012). December and January are the wettest months; about 80% of total annual precipitation occurs between November and April. The Troodos massif is the water tower of Cyprus and it covers more than 25% of the island's total area (9251 km$^2$). The 38 watersheds of the study form a radial drainage system around Troodos (Figure 1). A summary of the long-term average hydroclimatic characteristics of the studied watersheds is shown in Figure 1(a). The map of Cyprus with the boundaries of the 38 watersheds and the trends in precipitation and streamflow in the 1980-2015 period are also shown in Figure 1(b). For precipitation, Sen's slopes ranged from -2.2 to 3.2 mm·y$^{-1}$, with no statistically significant ($p < 0.05$) trends (Mann-Kendall test) among the 38 watersheds. For streamflow, Sen's slopes ranged from -0.22 to 0.01 Mm$^3$·y$^{-1}$ with statistically significant negative trends for two watersheds. The reference evapotranspiration (ET) trend (not shown) was statistically significant for all watersheds and Sen's slopes ranged from 1.3 to 3.8 mm·y$^{-1}$.



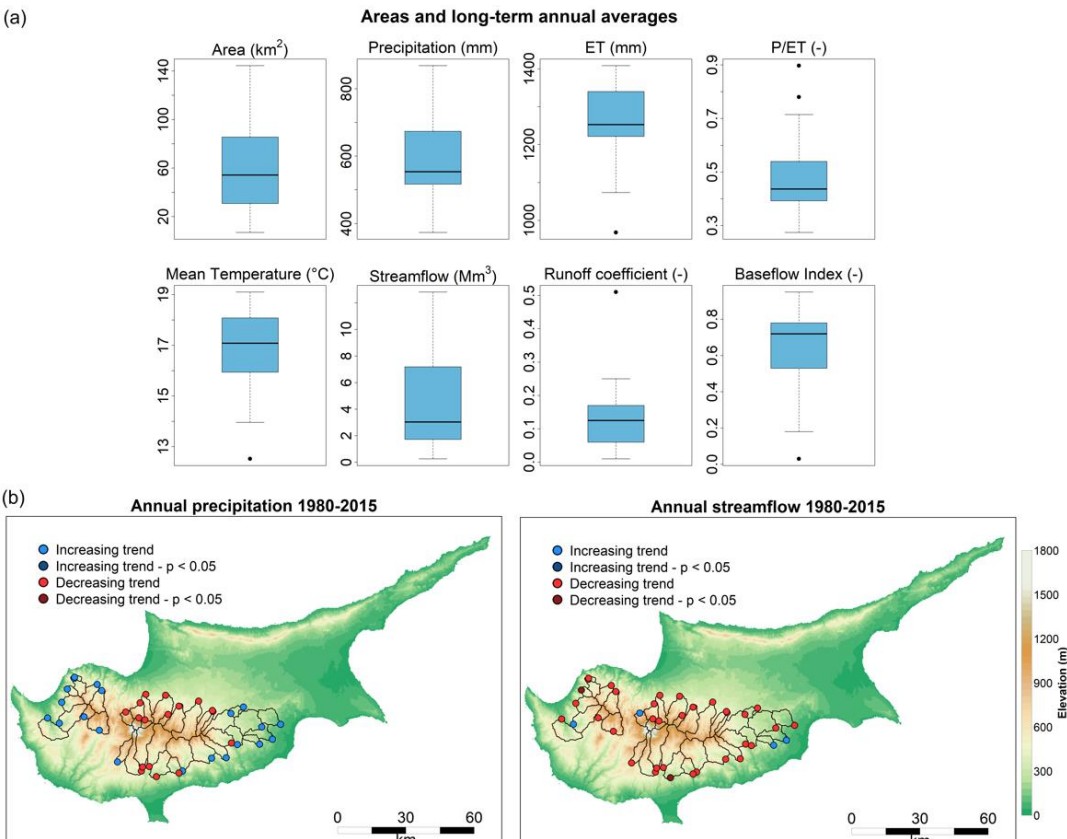

**Figure 1.** (a) Boxplots of the surface areas and the long-term hydrologic-year averages (1980-2015) of hydrological and climate variables of the 38 study watersheds; (b) Elevation map (m) of the island of Cyprus with the boundaries of the 38 watersheds and the trend and significance of trend, based on Mann-Kendall and Sen's slope statistics, for annual precipitation (left) and streamflow (right), for the 1980-2015 period, shown as circles at the outlets of the watersheds.

Daily precipitation and temperature observations for the area of Cyprus (CYOBS) for 1980-2015 in gridded datasets at 1 km spatial resolution were used for forcing the hydrological model (section 2.3) and the evaluation of RCM output (section 2.4). The periods 1980–1998 and 1998–2015 were used for calibration and validation, respectively. These gridded observations were generated with the spatial interpolation methods described by Camera et al. (2014) and Sofokleous et al. (2021). The temperature data include daily minimum and maximum values, from which the daily average temperature was computed. The daily reference evapotranspiration, required as input for the hydrological model was computed from the daily minimum and maximum temperature with the Hargreaves equation (Hargreaves & Samani, 1985). The daily data on watershed-level were extracted from the gridded 1-km data, by averaging the gridded data over the watershed areas. For the hydrologic model calibration and evaluation, daily streamflow observations at the outlets of the 38 watersheds from the monitoring network of the Water Development Department





of Cyprus (Figure 1), in the period 1980-2015, were used. Missing streamflow data (six watersheds, 4-14 years) were
filled, by fitting linear or exponential relationships with downstream or upstream stations in the same watershed. These
data were used for the calibration and evaluation of the hydrological model.

### 2.3 The GR4J conceptual hydrological model

The conceptual, four-parameter, daily GR4J model was used for the streamflow simulations. A detailed description
of the model is given by Perrin et al. (2003). The GR4J model has shown good performance for different environments
(e.g., Broderick et al., 2016; Le Coz et al., 2016; Guo et al., 2020). The four model parameters represent the watershed's
soil water or production storage (X1; mm), groundwater exchange acting on streamflow (X2; mm), streamflow storage
(X3; mm), and a time parameter for the unit hydrograph (X4; days). The groundwater exchange coefficient can add a
positive (inflow to stream) or negative (recharge losses) contribution to the water balance. The parameters are
embedded in exponential equations, which are used to represent the hydrological processes. All input data, i.e., the
forcing data of daily precipitation and reference evapotranspiration are expressed in mm over the watershed area. The
GR4J model implementation in R (Coron et al., 2023) is used. The parameter optimization in the model uses the
steepest descent local search algorithm. The GR4J model was applied on the 38 watersheds of the study area with
streamflow records for the hydrologic years 1980-81 to 2014-15 for model calibration and validation.


### 2.4 Selection of RCMs for the forcing of future streamflow simulations

Precipitation and daily minimum and maximum temperature were extracted for the domain of Cyprus from 18 RCMs
of the EURO-CORDEX (European hub of the international Coordinated Regional Climate Downscaling Experiment;
Jacob et al., 2020) in the past period (1980-2010) and the future period 2030-2060. This ensemble is based on a
combination of six driving GCMs and seven RCMs used for dynamical downscaling. The historical period 1980-2010
is used as the reference period of the study, to which future precipitation changes are computed. The RCM horizontal
spatial resolution is 0.11° (~12 km), and the temporal resolution is at the daily time step. The RCP8.5 "business-as-
usual" scenario (Meinshausen et al., 2011) was selected, in light of current global development pathways (Pedersen
et al., 2021).

The EURO-CORDEX data were bias-corrected and downscaled using the quantile delta mapping (QDM) algorithm
(Cannon et al., 2015). The QDM method preserves the relative (precipitation) or absolute (temperature) changes in
quantiles simulated by the RCM. The R MBC Package of Cannon (2024) was used for the QDM application. The
CYOBS area was covered by 55 land-based RCM grid cells. For the downscaling of the climate data from 12-km to
1-km resolution, the RCM values were re-gridded to the 1-km grid of the CYOBS data and the bias correction
algorithm was applied with the 1-km CYOBS and the re-gridded RCM data. To ensure that bias-corrected values of
daily minimum temperature ($T_{min}$) do not exceed those of daily maximum temperature ($T_{max}$), the diurnal temperature
range (DTR = $T_{max} - T_{min}$) and the $T_{max}$ were bias-corrected, following the suggestion by Thrasher et al. (2012), who
found that bias-correcting $T_{max}$ and DTR resulted in smaller errors.





The 18 bias-corrected and downscaled RCMs were evaluated for their skill in reproducing observed precipitation
220 indices and for their skill in reproducing observed streamflow, when these RCMs are used as forcing for the calibrated
GR4J model in the 1980-2010 reference period. The precipitation indices were computed as long-term averages of
annual index values. These are (i) the average and (ii) the standard deviation of annual precipitation, (iii) the ratio of
the precipitation of the five wettest months to the annual precipitation (W5R; from November to March), (iv) the
Simple Daily Intensity Index (SDII), which is the average precipitation rate on wet days (≥1mm) and (v) the number
225 of days with daily precipitation exceeding 10 mm (R10mm). For streamflow, the relative error of the simulated to
observed annual streamflow in each watershed was computed. Based on this evaluation, a subset of RCMs best
simulating the reference period precipitation and streamflow were selected to be used for modeling the future
streamflow with the calibrated GR4J parameterization.

230 **2.5 Future projections of water resources**

The changes in future precipitation and streamflow, relative to the reference period, were evaluated for the total period
changes and the changes in the driest and wettest years, i.e., in the two consecutive driest and wettest years as well as
the five driest and five wettest years overall. The changes in precipitation, reference evapotranspiration (ET) and
streamflow were also analyzed for groups of the 38 watersheds, defined by four flow regime types. The four flow
235 regime types are:  Permanent flow (P), Intermittent pools flow (I-p), Intermittent harsh flow (I-h), and Ephemeral flow
(E). These types of flow regimes are derived from two indices of monthly streamflow data, including the flow
permanence (Mf), i.e., the long-term mean annual relative number of months with flow, and the six-month seasonal
predictability of dry periods (SD6), based on the ratio of multi-annual frequencies of the zero-flow months for the
contiguous six wetter months of the year to the frequencies of the zero-flow months for the contiguous six drier
240 months. A detailed description of the derivation of the two indices and the flow regime type is given in Gallart et al.
(2012).The flow regime type for the reference period was assigned by the Water Development Department of Cyprus
and the future flow regime type was computed from the simulated flow, modelled with the climate projections of the
different RCMs.

**3 Results**

245 **3.1 Selection of objective function under changing climate conditions**

The matrix of evaluation of the six objective functions based on the median values of the 38 watersheds (averaged for
all 5-year periods) for each performance measure is shown in Table 1. In the calibration runs, the best evaluation
measure values were achieved when the evaluation measure was used as the objective function, shown in bold in the
table. These results corroborate that the optimization routine in the GR4J model implementation in R performs
250 reasonably well.

The CSS, which normalizes the evaluation measure values and combines them into one relative performance score,
shows that KGE is the best in the calibration period for 31 out of 38 watersheds, followed by NSEsqrt. Under the
conventional split-sample method, the model parameterization obtained with the KGE (0.89) in the calibration would





be selected and the validation, based on the full set of 182 5-year periods (14 5-year calibrations x 13 5-year
validations), would confirm an acceptable performance (KGE 0.53). However, the evaluation matrices show that the
parameterization obtained with the NSEsqrt in the calibration received higher scores for the validation (NSEsqrt 0.74,
KGE 0.59) than the KGE parameterization (NSEsqrt 0.70, KGE 0.53). In the validation, NSEsqrt also outperformed
the other objective functions in 14 watersheds, whereas the KGE parameterization outperformed the other objective
functions in three watersheds only, which is the second worst performance out of the six functions. Interestingly, the
NSE-based functions led to better model performance, based on CSS, for a larger number of watersheds in the
validation compared to the KGE-based functions. The NSE-based functions also achieved a much smaller PBIAS than
the KGE-based functions in the validation period, opposite to the lower PBIAS with KGE-based functions in the
calibration.

The comparison of the objective functions, based on the CSS and across different long-term average precipitation of
the 38 watersheds in Figure 2, shows that the optimization done with NSEsqrt is on average the best for all precipitation
regimes in the validation for the 182 5-year validation periods, whereas the CSS based on KGE has no important
difference from the NSE, NSElog and KGElog functions. KGEsqrt is consistently performing worse. This surprising
contrast of KGE performance in the validation relative to the calibration compared to the NSE-based function on
NSEsqrt, shows that the square root form of the NSE is a more skillful objective function than KGE formulation in
the model evaluation in independent evaluation data periods.

Subselecting calibration-validation periods with specific changes in climate conditions from the calibration to the
validation reveals some model performance losses or gains (Table 1). Both wetter validation periods (ΔP>15% and
5%<ΔP≤15%) achieved comparable performance with the validation period with small changes in precipitation (-
5%<ΔP≤5%), whereas the evaluation of the drier validation (ΔP≤-5%) showed the worst performance compared to
the other three classes. This is evident in the reduction of the values of the six goodness-of-fit measures and the
increase in the PBIAS values, indicating that the observed hydrographs and the total streamflow volumes are simulated
with less accuracy in drier validation periods than in wetter validation periods. NSEsqrt led to the lowest increase, on
average, in PBIAS in the driest validation periods, i.e., with a PBIAS of 9%, compared to the other objective functions,
where PBIAS ranged from 17% to 39%. The comparison of the objective functions performance based on CSS for the
wettest validation periods (Figure 2) shows the close distance of the objective functions across all precipitation regimes
of the 38 watersheds. For the drying condition in validation, NSEsqrt stands out as the best-performing objective
function across all precipitation regimes, with also increasing distance from the other functions for the wettest
watersheds.

In the overall comparison of the six objective functions with the use of the unitless and objective CSS, the KGE ranks
first (0.74) in the calibration runs, followed by NSEsqrt (0.65). In the full set of 182 validation runs, NSEsqrt ranks
first (0.66), followed by NSElog (0.62), KGE (0.60) and NSE (0.59). NSEsqrt ranks also first in all four precipitation
change classes (CSS from 0.66 to 0.70), whereas KGE ranks second, or third and even fourth. Due to its consistent
performance for all climate change conditions, the objective function of NSEsqrt is selected for the model optimization
for the future streamflow simulations with the GR4J model.





**Table 1.** The median values (38 watersheds) of the seven evaluation measures and the CSS (columns; averaged over all 5-year calibration and validation periods per watershed) and the number of watersheds for which an objective function achieved the highest CSS relative to the other functions, for the simulated streamflow in the calibration and the four validations, for the model parameterizations obtained with each of the six objective functions (rows). In bold the best value for each evaluation measure (per column).

| Evaluation measures | NSE (-) | NSE$_{log}$ (-) | NSE$_{sqrt}$ (-) | KGE (-) | KGE$_{log}$ (-) | KGE$_{sqrt}$ (-) | PBIAS (%) | CSS (-) | #watersheds with max CSS (0-38) |
|---|---|---|---|---|---|---|---|---|---|
| Objective function | | | CALIBRATION (14 5-year periods) | | | | | | |
| NSE | **0.82** | 0.45 | 0.79 | 0.74 | 0.67 | 0.64 | -6 | 0.54 | 1 |
| NSElog | 0.63 | **0.75** | 0.78 | 0.68 | 0.82 | 0.65 | -6 | 0.60 | 1 |
| NSEsqrt | 0.74 | 0.64 | **0.84** | 0.78 | 0.75 | 0.81 | -9 | 0.65 | 3 |
| KGE | 0.78 | 0.58 | 0.81 | **0.89** | 0.76 | 0.75 | **0** | **0.74** | **31** |
| KGElog | 0.55 | 0.72 | 0.78 | 0.71 | **0.86** | 0.70 | 6 | 0.60 | 1 |
| KGEsqrt | 0.59 | 0.47 | 0.81 | 0.69 | 0.69 | **0.90** | -2 | 0.31 | 1 |
| | | | VALIDATION (182 5-year periods) | | | | | | |
| NSE | **0.51** | 0.35 | 0.71 | 0.53 | 0.58 | 0.58 | **-1** | 0.59 | 6 |
| NSElog | 0.45 | **0.62** | 0.67 | 0.51 | **0.69** | 0.49 | 8 | 0.62 | 9 |
| NSEsqrt | 0.50 | 0.49 | **0.74** | 0.59 | 0.68 | **0.70** | 4 | **0.66** | **14** |
| KGE | 0.39 | 0.45 | 0.70 | 0.53 | 0.67 | 0.59 | 15 | 0.60 | 3 |
| KGElog | 0.19 | 0.60 | 0.63 | 0.47 | 0.69 | 0.55 | 30 | 0.56 | 6 |
| KGEsqrt | 0.6 | 0.35 | 0.67 | 0.41 | 0.64 | 0.69 | 13 | 0.39 | 0 |
| | | | VALIDATION (18 five-year periods with ΔP>15%) | | | | | | |
| NSE | **0.53** | 0.39 | 0.72 | 0.55 | 0.58 | 0.63 | -6 | 0.54 | 4 |
| NSElog | 0.52 | **0.68** | 0.70 | 0.59 | **0.72** | 0.51 | **1** | 0.62 | 9 |
| NSEsqrt | **0.53** | 0.58 | **0.76** | **0.61** | 0.68 | **0.73** | -2 | **0.66** | **11** |
| KGE | 0.38 | 0.55 | 0.72 | 0.55 | 0.68 | 0.69 | 14 | 0.62 | 5 |
| KGElog | 0.44 | 0.64 | 0.69 | 0.54 | **0.72** | 0.57 | 18 | 0.61 | 8 |
| KGEsqrt | 0.07 | 0.44 | 0.69 | 0.42 | 0.63 | 0.73 | 16 | 0.42 | 1 |
| | | | VALIDATION (44 five-year periods with 5%<ΔP≤15%) | | | | | | |
| NSE | 0.51 | 0.48 | 0.74 | 0.58 | 0.61 | 0.64 | -3 | 0.60 | 5 |
| NSElog | 0.50 | **0.67** | 0.72 | 0.58 | **0.73** | 0.54 | 5 | 0.62 | 9 |
| NSEsqrt | **0.52** | 0.57 | **0.77** | **0.61** | 0.69 | **0.73** | **1** | **0.67** | **16** |
| KGE | 0.43 | 0.55 | 0.73 | 0.58 | 0.67 | 0.67 | 11 | 0.62 | 3 |
| KGElog | 0.36 | 0.66 | 0.70 | 0.52 | **0.73** | 0.60 | 18 | 0.55 | 4 |
| KGEsqrt | -0.05 | 0.45 | 0.69 | 0.45 | 0.64 | 0.74 | 8 | 0.38 | 1 |
| | | | VALIDATION (35 five-year periods with -5%<ΔP≤5%) | | | | | | |
| NSE | **0.55** | 0.40 | 0.71 | 0.55 | 0.62 | 0.57 | **1** | 0.61 | 7 |
| NSElog | 0.47 | **0.64** | 0.67 | 0.51 | **0.70** | 0.50 | 11 | 0.60 | 6 |
| NSEsqrt | 0.51 | 0.50 | **0.75** | 0.59 | 0.69 | 0.69 | 5 | **0.68** | **14** |
| KGE | 0.41 | 0.47 | 0.71 | 0.57 | 0.67 | 0.58 | 20 | 0.60 | 5 |
| KGElog | 0.23 | 0.60 | 0.64 | 0.46 | **0.70** | 0.56 | 33 | 0.54 | 6 |

none




| | | | | | | | | | |
|---|---|---|---|---|---|---|---|---|---|
| KGEsqrt | -0.05 | 0.41 | 0.69 | 0.41 | 0.65 | **0.69** | 15 | 0.40 | 0 |
| VALIDATION (22 five-year periods with ΔP≤-5%) | | | | | | | | | |
| NSE | 0.46 | 0.19 | 0.61 | 0.48 | 0.56 | 0.48 | 20 | 0.59 | 5 |
| NSElog | 0.30 | **0.55** | 0.56 | 0.43 | **0.68** | 0.45 | 26 | 0.61 | 7 |
| NSEsqrt | **0.48** | 0.40 | **0.68** | **0.54** | 0.64 | 0.60 | **9** | **0.70** | **21** |
| KGE | 0.32 | 0.35 | 0.63 | 0.48 | 0.64 | 0.47 | 20 | 0.57 | 1 |
| KGElog | 0.06 | 0.54 | 0.54 | 0.32 | 0.67 | 0.49 | 39 | 0.55 | 4 |
| KGEsqrt | -0.29 | 0.29 | 0.61 | 0.38 | 0.59 | **0.62** | 17 | 0.41 | 0 |


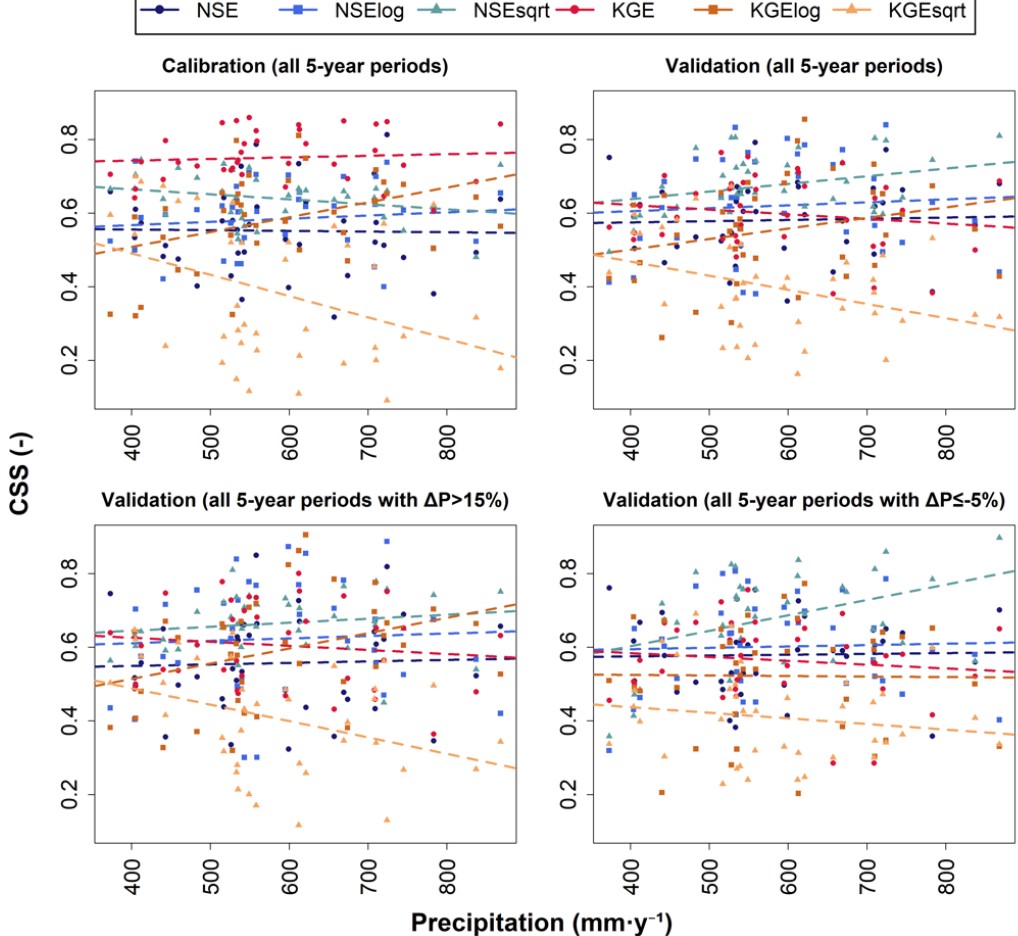

**Figure 2.** Scatterplots with the composite scaled score (CSS) of the parameterizations obtained with each the six objective functions for 38 watersheds (y-axis), plotted against the long-term average annual precipitation of the watersheds (x-axis). The dashed lines represent the linear trend between CSS and precipitation per objective function.




### 3.2 Selection of model parameterization for future simulations

To select the optimal GR4J parameter set for the simulation of streamflow in the future period, the NSEsqrt values of the 14 5-year calibration windows and their respective NSEsqrt values in the five validation periods were extracted per watershed. Figure 3 shows the best NSEsqrt value of the 14 5-year calibration runs and the best average NSEsqrt value of the validation of each of these 14 model parameterizations over the 13 validation runs (all), and similarly over the validation runs for each of the four precipitation change classes. These best NSEsqrt values identify the best performing model parameterization (out of the 14) for different climate conditions, for each watershed. The median and interquartile range of the NSEsqrt values are also shown. For the future simulation experiments, the parameter set from the 5-year calibration run, with the highest validation performance in drying conditions, i.e., $-5\% < \Delta P \leq 5\%$ and $\Delta P \leq -5\%$, in line with the climate model projections for the study area, was selected for each watershed.

Interestingly, nearly all watersheds were simulated with acceptable performance by all calibration sets, with the best and median NSEsqrt of the 14 parameterizations on average equal to 0.88 and 0.83 for the calibration and 0.78 and 0.69 for all validation runs, respectively. The average interquartile range of NSEsqrt obtained with the 14 parameterizations, equal to 0.09 for the calibration and 0.15 for any validation experiment for all watersheds, shows the small variability in the performance of GR4J with most of the 14 parameterizations based on the 5-year calibration periods.

Examining the same results, with the watersheds compared by their long-term average precipitation, it is seen that lower NSEsqrt values, by about 0.1 and greater variability, up to 0.2, among the 14 parameterizations are obtained for the ten driest watersheds (average precipitation 373 to 517 mm·y$^{-1}$) compared to the ten wettest watersheds (average precipitation 674 to 868 mm·y$^{-1}$). The 12$^{th}$ watershed (Gialia near Pano Gialia, 528 mm·y$^{-1}$ precipitation) stands out as the worst performing out of the 38 watersheds in all validation sets with a median NSEsqrt below 0.2. This particular watershed is only 16 km$^2$ in size and has a small dam upstream the streamflow gauge of the watershed. The best performing watershed in all validation sets (Stavros tis Psokas, 621 mm·y$^{-1}$), with median NSEsqrt above 0.89 has a size of 86 km$^2$, which classifies the watershed as medium to large-sized (Figure 1).


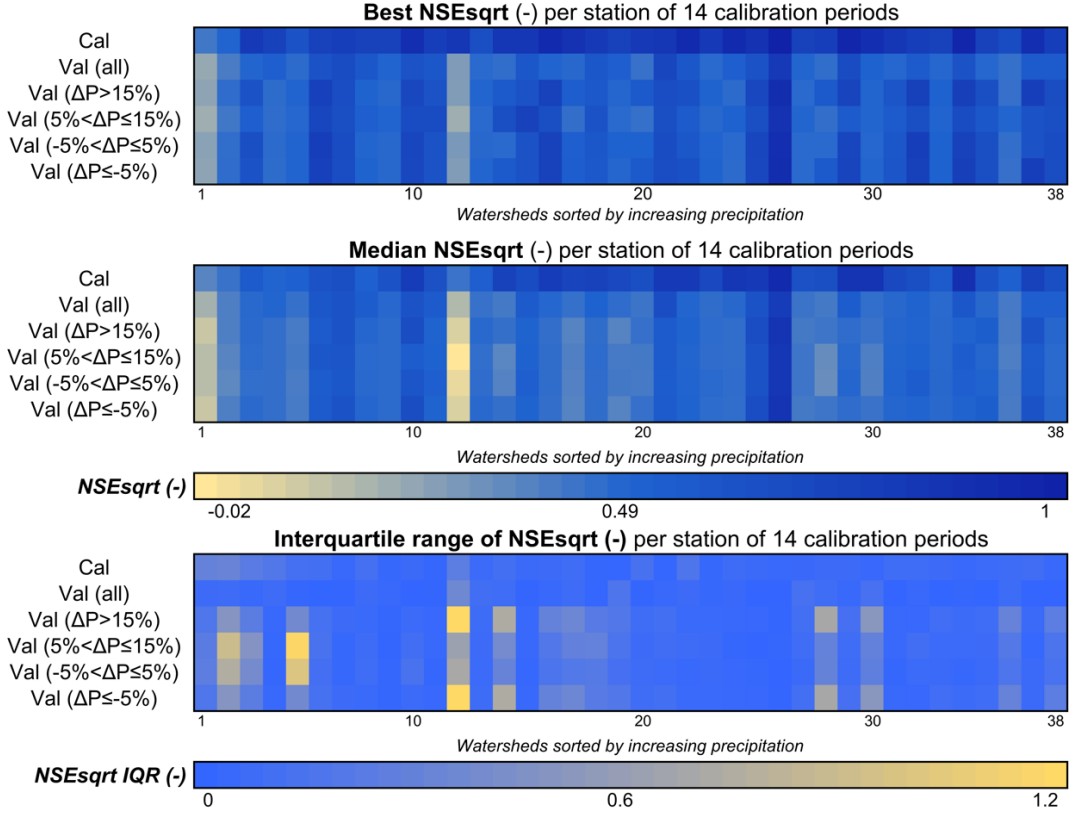

**Figure 3** Heatmap with the best (top), median (middle) and interquartile range (IQR) (bottom) of NSEsqrt (-) for each watershed, sorted by increasing long-term annual average precipitation (columns) and each calibration-validation set of runs (rows). The values of NSEsqrt or the IQR for each cell in the heatmap are computed as averages of the values from the 14 5-year calibration runs, the 13 validation runs for each of the 14 calibration runs. i.e., Val (all), and the validation runs corresponding to each of the four precipitation change classes, as specified in Table 1.

**3.3 Bias correction and selection of RCMs**

The bias correction of the RCMs with the QDM method yielded model indices very close to the observed values for total precipitation, with a maximum difference of 3% and for the SDII and R10mm with minimal difference from the observed values for all 18 RCMS (Table 2). The top nine models listed in Table 2 comprise the subset of RCMs used to quantify future changes in streamflow. These nine models simultaneously achieved the lowest error in the simulation of the ratio of the precipitation in the five wettest months to the annual total (W5R) and the simulation of annual average streamflow. W5R ranged between 0.79 and 0.84 for the selected RCMs and between 0.60 and 0.73 for the other nine models, compared to the observed 0.82 W5R value. The relative error in the simulation of streamflow





ranged between 1% and 11% in absolute values for the selected RCMs and between 5% and 40% for the remaining models. The W5R, as an evaluator of bias-corrected precipitation, has the strongest correlation (0.86) with the error
in streamflow simulations compared to the correlation of streamflow error with any of the other four precipitation indices (0.13-0.25).

**Table 2**. Indices of the observed (CYOBS) and RCM-simulated precipitation for the 1980-2010 reference period for 18 RCMs before and after bias correction for the full area of the Republic of Cyprus (Section 2.2); and streamflow bias of the GR4J simulations with the RCM forcing relative to observed streamflow for the area of the 38 watersheds.
The Pearson's correlation coefficient, denoting the correlation between each precipitation index and the streamflow bias for the 18 RCMs is shown on the last line. In bold font, the nine models with W5R less than 10% difference from the observed W5R and streamflow relative error less than 15%.

| Dataset | Precipitation | | | | | Precipitation | | | | | Streamflow |
|---|---|---|---|---|---|---|---|---|---|---|---|
| | Average annual | St. dev.[1] | W5R[1] | SDII[1] | R10mm[1] | Average annual | St. dev. | W5R | SDII | R10mm | Average annual |
| | (mm/y) | (mm/y) | | (mm/d) | (days/y) | (mm/y) | (mm/y) | | (mm/d) | (days/y) | ($10^6$m$^3$) |
| CYOBS | 467 | 93 | 0.82 | 6.1 | 14 | 467 | 93 | 0.82 | 6.1 | 14 | 173 |
| | *Raw RCM data* | | | | | *Bias corrected RCM data (QDM)* | | | | | Relative error[2] |
| EC-EARTH_HIRHAM | 452 | 90 | 0.84 | 5.1 | 13 | 477 | 96 | **0.84** | 6.2 | 14 | **-0.01** |
| NorESM_RCA | 609 | 128 | 0.75 | 4.9 | 19 | 460 | 122 | **0.80** | 6.0 | 14 | **-0.02** |
| NorESM_RACMO | 455 | 110 | 0.77 | 3.6 | 13 | 466 | 132 | **0.80** | 6.1 | 14 | **0.04** |
| HADGEM_RACMO | 562 | 112 | 0.76 | 3.8 | 16 | 474 | 107 | **0.80** | 6.2 | 14 | **0.05** |
| HADGEM_HIRHAM | 422 | 118 | 0.82 | 4.1 | 12 | 474 | 134 | **0.83** | 6.1 | 14 | **0.07** |
| MPI_RACMO | 509 | 100 | 0.79 | 3.8 | 14 | 452 | 103 | **0.83** | 6.0 | 14 | **-0.08** |
| EC-EARTH_RACMO | 459 | 75 | 0.79 | 3.8 | 13 | 472 | 86 | **0.82** | 6.2 | 14 | **-0.09** |
| EC-EARTH_RCA | 596 | 85 | 0.76 | 5.2 | 18 | 474 | 78 | **0.80** | 6.2 | 14 | **-0.09** |
| MPI_RCA | 617 | 155 | 0.75 | 5.0 | 19 | 461 | 141 | **0.79** | 6.0 | 14 | **-0.11** |
| HADGEM_RCA | 687 | 158 | 0.68 | 5.1 | 20 | 479 | 136 | 0.73 | 6.2 | 15 | -0.05 |
| CNRM_RACMO | 517 | 111 | 0.67 | 3.5 | 14 | 470 | 127 | 0.71 | 6.1 | 14 | -0.11 |
| CNRM_ALADIN | 840 | 149 | 0.67 | 6.0 | 27 | 476 | 112 | 0.70 | 6.2 | 14 | -0.17 |
| EC-EARTH_CLM | 320 | 87 | 0.71 | 4.0 | 9 | 465 | 124 | 0.71 | 6.1 | 14 | -0.19 |
| MPI_REGCM | 615 | 86 | 0.68 | 3.4 | 14 | 468 | 92 | 0.71 | 6.1 | 14 | -0.31 |
| MPI_REMO | 476 | 98 | 0.75 | 4.2 | 12 | 454 | 88 | 0.70 | 6.0 | 14 | -0.38 |
| HADGEM_REGCM | 561 | 78 | 0.64 | 2.9 | 11 | 479 | 96 | 0.67 | 6.2 | 14 | -0.39 |
| NorESM_REMO | 386 | 101 | 0.66 | 4.1 | 10 | 463 | 117 | 0.60 | 6.0 | 14 | -0.39 |
| IPSL_REMO | 275 | 83 | 0.69 | 3.5 | 7 | 474 | 129 | 0.61 | 6.1 | 14 | -0.40 |
| Pearson's r | 0.21 | 0.39 | 0.67 | 0.32 | 0.39 | 0.13 | 0.25 | 0.86 | 0.23 | 0.15 | |

[1] Standard deviation (St.dev); Ratio of precipitation of the five wettest months to the annual precipitation (W5R);
Simple daily intensity index (SDII); Number of days with daily precipitation exceeding 10 mm (R10mm)



[2] Relative error of modeled streamflow with RCM forcing relative to observed streamflow: (Qrcm.past-Qobs)/Qobs

### 3.4 Future projections of water resources

The 2030-2060 total streamflow of the 38 watersheds is projected to remain the same in the best model case or decrease up to 39%, in the worst case, relative to the 1980-2010 reference period value of 173 $Mm^3 \cdot y^{-1}$. The median projected change in streamflow was a reduction of 17%, and it was given by the NorESM-RACMO RCM. The change in precipitation could reach 16% in the driest model and 6% in the median model, relative to the 562 $mm \cdot y^{-1}$ over the 38 watersheds for the reference period (Table 3). The projected change in streamflow magnitude (14%-39%) was nearly double or more than double the change in precipitation (6%-16%) for six out of the nine models that provide the largest changes.

To understand the implications for water resources management, such as dam storage and irrigation supply, Table 3 shows the analysis of the 30-year annual series of precipitation for the driest years in the reference period and the driest years projected in the future period. According to these results, the driest 2-year period in the future could be from 3% up to 38% drier that the historical reference, with a median change of 11%. Streamflow in the two driest years could be 9% to 70% less in the future, with a median value of 36%. The top-5 driest years could have, on average, a median reduction in precipitation of 17% and 35% in streamflow. These reductions during these driest future periods are significantly greater than the projected 30-year long-term reductions, indicating that dry years in future will cause even more significant drought conditions than in the past.

A similar analysis for the wettest 2-year period and the top-5 wettest years shows a small increase in precipitation, with a median value of 15% and 4%, respectively, and nearly no change in streamflow relative to the streamflow of the wettest years of the past. These results indicate that, while very wet conditions are also expected in the future in some years, as in the past, the magnitude of reduction of water resources in the driest years in the future is disproportionally larger than the positive change in the wettest years.





**Table 3.** Annual averages of observed precipitation (CYOBS) and streamflow of the 38 watersheds for 1980-2010, driest and wettest 2 year periods, and top 5 driest and wettest years, and relative changes for 2030-2060 versus 1980-2010, for nine RCMs.

|  |  | 30-year average annual total | Driest 2-year period | Top 5 driest years | Wettest 2-year period | Top 5 wettest years | # watersheds with change in flow regime type |
|---|---|---|---|---|---|---|---|
| CYOBS | **Precipitation[1]** (mm·y$^{-1}$) | 562 | 383 | 393 | 720 | 730 | - |
| MPI-RACMO |  | -0.02 | -0.03 | -0.07 | 0.18 | 0.22 |  |
| EC-EARTH-RACMO |  | -0.01 | -0.10 | -0.08 | 0.07 | 0.06 |  |
| MPI-RCA |  | 0.00 | -0.34 | -0.20 | 0.04 | 0.15 |  |
| EC-EARTH-RCA | Relative precipitation change | -0.07 | -0.11 | -0.09 | 0.20 | 0.03 |  |
| NorESM-RACMO |  | -0.06 | -0.38 | -0.22 | 0.28 | 0.05 | - |
| HADGEM-RACMO |  | -0.09 | -0.05 | -0.11 | 0.15 | 0.00 |  |
| NorESM-RCA |  | -0.06 | -0.09 | -0.17 | 0.01 | 0.04 |  |
| EC-EARTH-HIRHAM |  | -0.16 | -0.22 | -0.24 | 0.22 | -0.06 |  |
| HADGEM-HIRHAM |  | -0.16 | -0.15 | -0.21 | 0.10 | -0.01 |  |
| Median |  | -0.06 | -0.11 | -0.17 | 0.15 | 0.04 | - |
| OBS | **Streamflow** (Mm$^3$·y$^{-1}$) | 173 | 53 | 51 | 348 | 345 |  |
| MPI-RACMO |  | 0.06 | -0.09 | -0.15 | 0.30 | 0.52 | 5 |
| EC-EARTH-RACMO |  | 0.03 | -0.36 | -0.08 | 0.08 | 0.06 | 3 |
| MPI-RCA |  | 0.00 | -0.61 | -0.32 | -0.01 | 0.21 | 4 |
| EC-EARTH-RCA | Relative streamflow change | -0.14 | -0.28 | -0.32 | 0.02 | 0.02 | 12 |
| NorESM-RACMO |  | -0.17 | -0.34 | -0.50 | 0.09 | -0.03 | 8 |
| HADGEM-RACMO |  | -0.25 | -0.43 | -0.35 | 0.00 | -0.05 | 6 |
| NorESM-RCA |  | -0.28 | -0.36 | -0.46 | -0.24 | -0.14 | 7 |
| EC-EARTH-HIRHAM |  | -0.36 | -0.70 | -0.55 | -0.05 | -0.22 | 10 |
| HADGEM-HIRHAM |  | -0.39 | -0.70 | -0.66 | 0.05 | -0.12 | 9 |
| Median |  | -0.17 | -0.36 | -0.35 | 0.02 | -0.03 | 7 |

[1] Precipitation from the CYOBS dataset refers to the precipitation over the 38 watershed areas, which is different from the CYOBS precipitation over the area of the island in Table 2.

The future change in precipitation and reference evapotranspiration and the change in streamflow for the four flow type classes of the 38 watersheds are shown in Figure 4(a). Results are presented for the three models with the minimum, median and maximum streamflow projected change among the nine RCMs. The HADGEM-HIRHAM model simulates the maximum streamflow change in the future, from -12% to -47%, for the median changes of each of the four flow types. The same model also projects the highest precipitation reduction (11%-17%) and ET increase (7%-9%) among the three models. For the median model, the NorESM-RACMO, with streamflow change up to 20% in the future for the four flow types, the ET increase is the least among the three models and equal to about 4%.



Overall, the streamflow changes in the future from different RCMs have a variability attributed to the variability of both precipitation and ET.

The watersheds with permanent flow (P) and intermittent pool (I-p) flow regime, which contribute 71% in the total
405 volume of streamflow of the 38 watersheds, are projected to exhibit the highest future reduction in streamflow, up to -47% and -42% respectively, relative to the reduction up to 20% for intermittent harsh (I-h) and up to 12% for ephemeral streams (E) according to HADGEM-HIRHAM RCM in Figure 4(a). The median model, NorESM-RACMO and the model with the least projected streamflow change, MPI-RACMO, also show that P and I-p flow types will be subjected to the highest reductions relative to I-h and E flow regimes. The flow regime type will also become subject
410 to change, implying a negative impact on the ecological conditions in the area of the 38 watersheds. Figure 4(b) shows that two watersheds with P, three with I-p and one with I-h flow regime have the highest probability (five or more models out of the nine) for a shift in their flow regime, while another one watershed with P, eight watersheds with I-p are and three with I-h are also projected to exhibit a change in flow regime, according to fewer than five out of the nine models. Overall, 18 out of the 38 watersheds are indicated to exhibit a change in their flow regime according to
415 at least one of the nine RCMs.

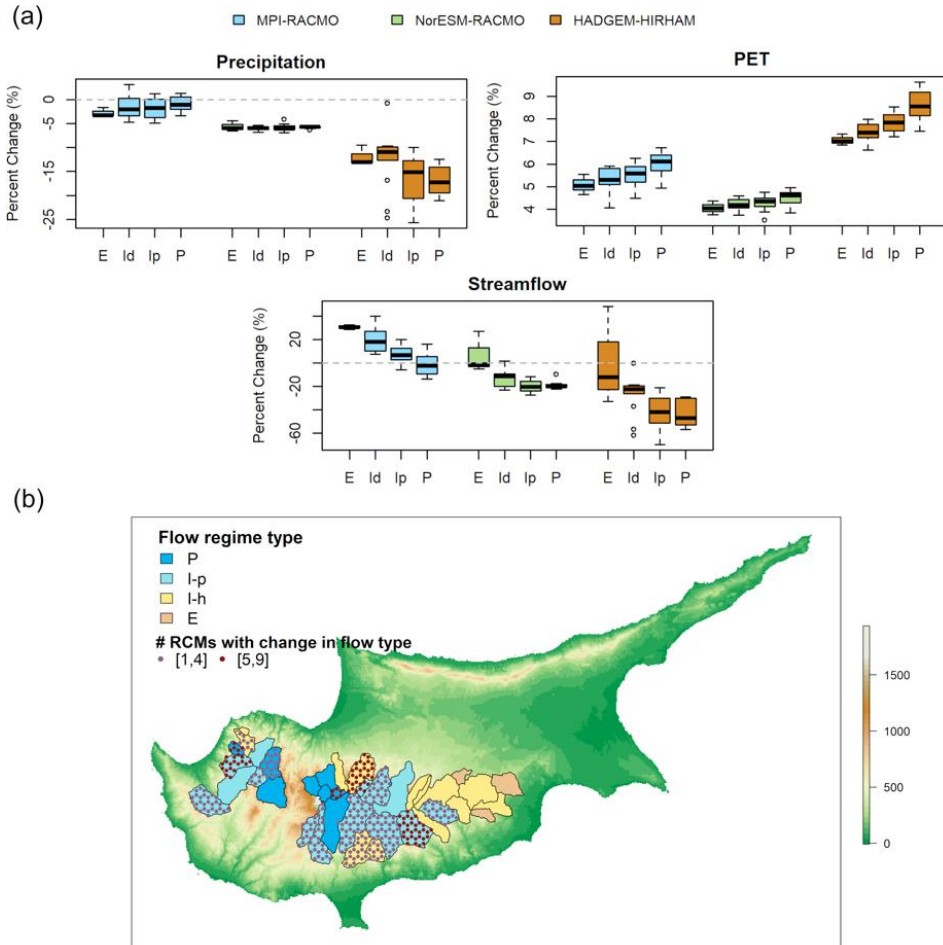

**Figure 4.** (a) Boxplots for the per cent changes in annual precipitation, reference evapotranspiration (ET) and streamflow in the 2030-2060 period relative to the 1980-2010 period simulated by three RCMs (corresponding to minimum, median and maximum streamflow change among nine RCMs) for 38 watersheds grouped by their flow regime type (E-Ephemeral, three watersheds, Ih – Intermittent harsh, 13 watersheds, Ip – Intermittent pools, 14 watersheds, P – Permanent, eight watersheds). (b) Elevation map of Cyprus (m) with the flow regime type of the 38 watersheds in the reference period (1980 -2010) and the number of models out of the nine RCMs projecting a change in the flow type in the future period (2030-2060).



## 4 Discussion

### 4.1 Selection of objective function and model parameterization under changing climate conditions

This study examined the changes in the hydrological model performance when the model is used at contrasting climate periods with multiple performance measures used for the calibration and validation. Although the Differential Split Sample Test approach has been used by several studies in the context of model transferability, with different lengths in the examined sub-periods, the transferability performance was evaluated with different or fewer number of measures than those used for the model calibration. For instance, Dakhlaoui et al. (2017; 2019) used 7-8 year long sub-periods or sub-periods comprised of discontinuous years with KGE as the calibration measure and NSE and Volume error as the transferability evaluators. Guo et al. (2020) used the KGE for calibration of sub-periods separated by 3-month windows and validated the model using KGE and the three components of KGE. The matrix-based evaluation of six objective functions in the current study provides a comprehensive evaluation of how KGE and NSE, with no transformation, the logarithmic and the square transformation of streamflow values perform during calibration and validation under contrasting climate conditions. KGE was found to be the optimal measure for model optimization, for the calibration period only. It ranked second in the validation periods with increased wetness ($\Delta P \geq 15\%$) and third or fourth in the full validation set and the validations with a small change in precipitation or increasing dryness (-5%$<\Delta P \leq$5% and $\Delta P \leq$-5%). This outcome for the 5-year calibration runs of the study with KGE, a widely used objective function and evaluation measure, could be further investigated by assessing the model calibration and validation performance using the three components of KGE as objective functions, separately. Fowler et al. (2018) found that a split-KGE, i.e., computed individually per year and averaged, performed better as an objective function than when KGE was computed for the full period.

The value of the composite score CSS, which normalizes and combines the values of seven evaluation measures, was highest for the NSEsqrt objective function. This finding agrees with Seiller et al. (2017), who found that the square root transformation with NSE is better, on average, for all flow regimes than calibration on NSE with inverse transformation or no transformation when the calibrated model is transferred to warmer conditions for catchments in southeastern Canada. These authors also showed that square root transformation is optimal at the expense of some loss of confidence for very high flows. In addition to the best relative performance of NSEsqrt for warmer conditions according to these authors, the current study showed that NSEsqrt was the optimal measure for warmer and wetter as well as for warmer and drier conditions in the validation. NSEsqrt ranked also first with increasing distance from the second highest ranked objective function with increasing precipitation reduction ($\Delta P$) in the validation period. This result suggests that NSEsqrt becomes more robust for increasing precipitation reduction in the validation period, relative to the calibration period. The comparison of the hydrological model performance for 14 GR4J parameterizations with NSEsqrt for the different validation experiments showed that drier watersheds, i.e., with lower precipitation than their wetter counterparts in the same Mediterranean study area, are simulated with lower values and higher variability of the performance measure, among the different parameterizations. These findings are in line with the study of Munoz-Castro et al. (2023), who found that the choice of the objective function was more important for high-aridity and lower runoff coefficients than for wetter climates and that a higher model performance and higher parameter agreement was achieved for wetter basins in their study of 92 basins in continental Chile.



Dakhlaoui et al. (2017) found that a 25% difference in precipitation and 1.75 °C in temperature were the thresholds for acceptable model transferability for streamflow simulations with changing climate conditions. The difference in climate conditions of the calibration-validation periods, used for the comparison of the objective functions in the present study, had a maximum temperature change of 1.5°C and precipitation change range from -14% to +25%, except one calibration-validation set with precipitation change of 32%.

**4.2 Bias correction and selection of RCMs**

The long-term average precipitation data from the EURO-CORDEX ensemble, evaluated before and after bias correction (Table 2), shows that this was an essential step for obtaining values close to the observations. The good performance of the Quantile Delta Mapping (QDM) method used here was highlighted in other studies (e.g., Meyer et al., 2019). However, 9 out of 18 RCMs showed poor reproduction of the seasonal cycle of precipitation. This leads to the conclusion that if an RCM fails to reasonably represent the seasonality of the reference period, it may also not be able to model future shifts. Mascaro et al. (2018) found that an ensemble of raw EURO-CORDEX RCMs underestimated winter precipitation in Sardinia, which has a similar climate and geographical extent as Cyprus. These authors' finding is in line with lower simulated precipitation in the five wettest months (WR5) by the RCMs in the current study, which was found to be correlated with the errors in simulated streamflow.

**4.3 Future projections of water resources**

The projected median change in future precipitation for 2030-2060 under RCP8.5, relative to 1980-2010, of the nine models (-6%, with a range from 2% to -16%), is in line with the overall observed drying of the island. These changes are within the range reported by other studies for the Mediterranean as a whole, about 10% in the first half and up to 20-40% in the second half of the 21st century (Zittis et al., 2021a). The projected median change of -17% for streamflow, with a range from 6% to -39% from the nine RCMs, indicates a pathway of deteriorating freshwater resources by the mid-21st century for Cyprus. Projected streamflow changes for two catchments on the southern slopes of the Troodos mountains by 2050 (up to 24% and 17%), as presented by Ragab et al. (2010), are within the projections of the current study.

Dakhlaoui et al. (2019) reported for 2040-2070, relative to 1970-2000, median runoff changes up to -6.2% under RCP4.5 (scenario not examined here) and from -13% to -31% under RCP8.5 for different RCMs and basins in Tunisia. For other Mediterranean locations and the end-of-the-century horizon, which was not analyzed here, runoff reductions seem to be close to 20% under RCP4.5, i.e. 19% in southern Italy (Senatore et al., 2022; reference period 1975-2005) and 16%-19% in Tunisia (Dakhlaoui et al., 2019). For the same horizon, relative to various 30-year periods in the second half of the 20th century, under the less optimistic, business-as-usual scenarios, i.e., B2 or RCP8.5, streamflow reductions will be exacerbated, i.e., 26%-54% in southern France (Lespinas et al., 2014), 50%-60% in central Spain (Sánchez-Gómez et al., 2023), and 37%-57% in Tunisia (Dakhlaoui et al., 2019).

The capacity of watersheds for streamflow generation will be weakened in future, given the larger relative change in streamflow than the relative change in precipitation. The difference in the magnitude of precipitation and streamflow projected changes are comparable to the differences in the magnitudes of changes of the two parameters computed from two different periods in the past (1916-1969 and 1970-2000), as found in a study for the islands' water resources



trends (Water Development Department, 2002). This difference in the magnitude of changes for these parameters can be partly attributed to the increased reference ET in the watersheds in the future (4%-8%), which is driven by the robust warming (Zittis et al., 2021b; Lazoglou et al., 2024). However, it is also known that rainfall-runoff relations are non-linear, because less runoff tends to be generated under drier watershed conditions. Additional stressors for streamflow generation are declines in groundwater levels, which are exacerbated by increased extractions for agricultural production in dry years (Zoumides et al., 2013; Leduc et al., 2017). Gutierrez et al. (2019) showed how the initial head in groundwater levels influences the onset of streamflow in Mediterranean climates. Previous modelling experience in Cyprus showed the sensitivity of simulated streamflow to the groundwater processes in hydrological modeling (Ragab et al., 2010; Camera et al., 2020; Sofokleous et al., 2023).

The change of flow regime type is most likely to occur for the wettest watersheds (Figure 6). This was also found by Reymond et al. (2019) who related the shift from a continuous flow regime to intermittent to the lower precipitation and increased number of consecutive days without rain. Pascual et al. (2015) showed that from the comparison of flow changes in three watersheds in northeastern Spain, the two wettest watersheds were expected to experience larger reductions in streamflow (34%) than the drier watershed (25%). Schneider et al (2012) found that the most remarkable flow regime alterations in Europe are expected in the Mediterranean. The results of this climate impact assessment on the fresh water resources in Cyprus contribute to the broadening of the knowledge of changes in streamflow at watershed level in the climate hotspot region of the eastern Mediterranean.

As shown with the range of changes for precipitation and streamflow presented above, uncertainty in future projections, particularly in climate change impacts on streamflow, is large. Given that the impacts of climate change are derived from a chain of modelling steps, the uncertainty in the future projections for streamflow could be identified for each step. In this study, multiple RCMs were used to drive the hydrological simulations. The emphasis was placed on multiple RCMs rather than multiple hydrological models, as previous research has shown that a significant source of uncertainty comes from the RCMs (Teutschbein and Seibert 2012). RCMs exhibit systematic biases in both temperature and precipitation, influenced by the individual structures of the GCMs and RCMs. For this reason, a bias correction step is necessary (Christensen et al., 2008). In this study, 18 different RCMs were selected, representing combinations of six GCMs downscaled by seven RCMs. A single hydrological model, GR4J, was used, as the particular model is a well-established model structure for streamflow simulations across a wide range of watershed conditions. Its performance and robustness have been found to be comparable to those of a multimodel approach (Seiller et al., 2017).

## 5 Conclusions

This study presented a method of objective function selection for hydrological model calibration for use in climate impact assessments. A matrix of six objective functions and seven evaluation measures and a composite scaled score (CSS), which normalizes and combines multiple evaluation measures into one score, is used to evaluate the hydrological model performance optimized with the six different objective functions over multiple calibration and validation sets for changing climate conditions.



The matrix method was applied to the GR4J hydrological model for 38 Mediterranean mountain watersheds, with average annual rainfall ranging between 373 and 868 mm and runoff coefficients between 0.01 and 0.51. The model parameterization obtained using the objective function of NSE with square root-transformed streamflow values (NSEsqrt) outperformed those obtained using NSE, NSElog, KGE, KGEsqrt, and KGElog during the validation period. The CSS showed that KGE outperformed the other functions only during the calibration period. The

parameterization optimized with NSEsqrt also resulted in the best model performance under four different types of changing climate conditions. Specifically, with increasing dryness in the climate conditions, NSEsqrt stood out as the best objective function, showing greater performance differences from the others than under wetter or unchanged conditions. NSEsqrt can therefore be used as an objective function for streamflow simulations in Mediterranean watersheds experiencing drying trends. The matrix method could also be applied to identify the most suitable objective

functions under changing climate conditions in other environments.

Optimal watershed calibrations were selected from the NSEsqrt results of the matrix method. In particular, 14 hydrological model parameterizations obtained using the selected objective function, from 14 respective five-year calibration runs, were evaluated in five-year validation runs across the full data period. They were also evaluated in the selected five-year validation runs that represented different changes in climate conditions relative to the calibration

period. For each of the 38 watersheds, the parameter set that performed best in validation windows with drying climate conditions, according to climate projections for the study region, was selected for the streamflow projections.

Comparison of validation periods with different precipitation changes relative to the calibration period showed higher total streamflow errors and degraded hydrological model performance, as described by NSE, KGE and different transformations of the two measures, for transitions to dry periods compared to the results for transitions to wet

periods. Likewise, the watersheds with lower annual precipitation showed lower and more variable performance measures compared to the wettest watersheds of the study area.

The evaluation of a large ensemble of RCMs showed that these models need to be bias-corrected, downscaled and evaluated for precipitation indicators and hydrological model performance when used for the assessment of climate impacts on fresh water resources. In particular, the comparison of the output of 18 RCMs and of the output of the

hydrological models forced by these RCMs revealed that correct simulation of seasonal distribution of precipitation is correlated with the hydrological model's ability to achieve low streamflow errors. This criterion may help increase confidence in an RCM's ability to simulate future changes in both seasonal precipitation patterns and streamflow.

For Cyprus, precipitation projections under RCP8.5 scenario showed median precipitation change of -6%, among nine RCMs, and a -16% change in the worst case for 2030-2060 period relative to the 1980-2010 reference period. The

streamflow reductions are amplified relative to the projected precipitation reductions. The projected median streamflow change is -17% and -39% in the worst case. The five driest years in future are expected to become even drier than the past, from 9% up to 70% drier in terms of total streamflow for 38 watersheds of the island. This outcome highlights the importance of examining projected changes in specific future years, as they may reveal acute water



shortages that must be considered in adaptation measures, something that long-term average analysis alone might
overlook.

Permanently flowing rivers were modelled to be affected the most by streamflow reductions and they have the highest
risk for a change in their flow regime, i.e. from permanent to intermittent. The changes in flow regime types imply
reduced or even absent water flow in streams during certain periods of the hydrological year, leading to declines in
ecosystems that depend on stream water. The pessimistic projections for the terrestrial water resources in 2030-2060
found in this study under the RCP8.5 scenario, or business-as-usual scenario in terms of emissions of greenhouse
gases, are comparable to the results found in other areas around the Mediterranean.

The study also showed the importance of maintaining long-term streamflow observations for watersheds with different
flow regime types. These observations are essential for parameterizing hydrological models and improving the land
surface models imbedded in climate models. Such models are needed to predict and project seasonal and future water
resources at both regional and watershed scales and to develop climate adaptation plans and water management
policies.

**Data and code availability**

The meteorological and streamflow data of the reference period can be requested from the Department of
Meteorology of Cyprus and the Water Development of Cyprus, respectively. The EURO-CORDEX data are
publicly available through the Earth System Grid Federation (https://esg-dn1.nsc.liu.se/) and the Copernicus Climate
Change Services Data Store (https://cds.climate.copernicus.eu). The climate data bias correction method is available
as an R package in the following link: https://cran.r-project.org/web/packages/MBC/index.html) and the
hydrological model is available as an R package in: https://hydrogr.github.io/airGR/

**Author contribution**

IS and AB designed the simulation experiments; GZ and GD collected the data; IS analyzed the data, performed the
simulations and wrote the manuscript; GZ, GD and AB reviewed and edited the manuscript.

**Competing interests**

The authors declare that they have no conflict of interest.

**Acknowledgments**

For the meteorological data of the reference period used in the study, we would like to thank our colleagues from the
Department of Meteorology of Cyprus. For the streamflow observations of the reference period, we would like to
thank our colleagues from the Water Development Department of Cyprus. For the RCM data, we would like to
acknowledge the CORDEX initiative. This research has received funding from the 3PRO-TROODOS Project



(INTEGRATED/0609/061), co-financed by the European Regional Development Fund and the Republic of Cyprus

through the Research and Innovation Foundation. This research was also supported by the PREVENT project, that

has received funding from the European Union's Horizon Europe Research and Innovation Program under Grant

Agreement No. 101081276.

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
