# Peer review of "Evaluating the Performance of Objective Functions and Regional Climate Models for Hydrologic Climate Change Impact Studies: A Case Study in the Eastern Mediterranean"

_EGUsphere, 2025_

## Author Comment (AC1)

**Response by the Authors to Reviewer 1 for EGUSPHERE-2025-2478**

We would like to sincerely thank the reviewer for his/her constructive review. Below, our detailed responses to each of the reviewer's comments. The original reviewer comments are presented in black font, and our responses are in blue font.

In this manuscript the authors demonstrate a robust hydrologically modelling method for simulating streamflow under projected future climate conditions.

While there is not anything particularly new in this manuscript the overarching method is well considered and supported by comprehensive modelling experiments. The manuscript is well structured, and the scientific literature well referenced throughout. Figures and tables are appropriate. Manuscript is generally well written but, in some cases, mixes tenses – benefit from further proofread to improve clarity.

Subject to revision this manuscript would make a useful addition to the scientific literature.

**Specific comments**

**Abstract**

Ln 12: Insert word ..... "conceptual" hydrological models.... Need to make it clear early that you are referring to conceptual hydrological models here. i.e. physically based models may not be compromised by non-stationary climate conditions.

Ln 14 Is "assess" the correct word here?

Ln 12 and 14 will be corrected to: "This study aims to (i) develop a method for selecting a skillful parameterization of a conceptual hydrological model under changing climate conditions and (ii) use the calibrated model to generate streamflow projections for 38 mountain watersheds in the eastern Mediterranean island of Cyprus over the next decades (2030-2060)."

Ln 18 here and later it is not clear to me how multiple 5-year windows between 1980-2015 resulted in 14 calibration and 182 validations. A little more explanation is required in the main body of manuscript, as it is not intuitive.

We will modify the first paragraph in Section 2.1.2 (Ln 126 - 130) as follows:

"Calibration was performed for all moving 5-year windows, within the calibration period (1980 – 1998), applying all six objective functions. Observational data and periods are given in Section 2.2. Validation was conducted for each calibration window and each corresponding calibrated model based on the six objective functions across 5-year windows within an independent validation period (1998 – 2015). The hydrologic year preceding each 5-year window was used as a warm-up year. In total, 182 (14 × 13) study model experiments resulting from 14 5-year calibration windows within 1980 and 1998 and 13 5-year validation windows within 1998 and 2015 were performed in this study. The objective functions were evaluated, firstly, for all 5-year calibration and 5-year validation periods corresponding to different changes in climate conditions...."

Ln 19 Matrix method. Reword, ambiguous.

Thank you. We will change the characterization of the suggested method to "comparative scheme", and instead of using "matrix", we use either "table" or no other word. Please see below some examples.

A matrix-based approach was developed -> comparative scheme
Based on the matrix method, the Nash-Sutcliffe Efficiency -> comparative method
The matrix-based evaluation of the six objective and .. -> "-"
This results in a matrix of six objective functions -> table
a summarizing 6x8 matrix, comprised of the median values of the 38 -> table
Optimal watershed calibrations were selected from the NSEsqrt results of the matrix
method -> comparative table scheme

Ln 24 "...used to simulation streamflow with GR4J..." reword to make it clear streamflow was simulated using GR4J using inputs from the RCMs.

Ln 24 will be changed to: "Eighteen Regional Climate Models (RCMs) were bias-corrected, downscaled to 1 km and used as forcing in GR4J to simulate streamflow for 1980-2010"

Ln 30 Here and elsewhere I don't really like the term deteriorate in this context. It is ambiguous. Could you be more specific e.g. mean annual streamflow will decrease.

Ln 30 will be changed to: "Our findings suggest that mean annual streamflow in the eastern Mediterranean may deteriorate decrease significantly in the coming decades."

Ln 483 will be changed to: "The projected median change of -17% for streamflow, with a range from 6% to -39% from the nine RCMs, indicates a pathway of deteriorating reducing freshwater resources by the mid-21st century for Cyprus."

Ln 69 Hageman et al. (2013) is more than 12 years old. Is there not a more recent study using CMIP models?

Ln 69 This reference will be replaced by: "Asadieh and Krakauer (2017), using biascorrected meteorological outputs of five GCMs from CMIP5 to project streamflow changes, showed that southern Europe and Middle East, with southern North America and the Southern Hemisphere, will experience strong decrease in all percentiles of streamflow, highlighting the drought hazard risks under RCP8.5 by the end of the century."

Asadieh, B., & Krakauer, N. Y. (2017). Global change in streamflow extremes under climate change over the 21st century. Hydrology and Earth System Sciences, 21(11), 5863-5874. https://doi.org/10.5194/hess-21-5863-2017

Ln 78 Which phase of the CMIP?

Ln 78 will be changed to: "Cos et al. (2022) found that simulations from two phases of the Coupled Model Intercomparison Project (CMIP5 and CMPI6, at 1° spatial resolution) project a stronger warming in the Mediterranean, relative to the global mean change, particularly in the summer, which could range from 1.8°C to an alarming 8.5°C by the end of the century."

Ln 80 RCP? So this manuscript is using CMIP5 models. It does beg the question how different CMIP5 is from CMIP6 over the Mediterranean region? Hopefully this is covered in the discussion as it would be necessary to place the results of this study into context with the latest climate modelling.

Thank you pointing this out. We will add the following discussion on this topic in the Discussion section 4.2: "Bias correction and selection of RCMs"

"Comparison of CMIP5 and CMIP6 models in previous global and regional studies showed that uncertainty in precipitation is high in both generations of models (Cos et al. 2022; Wu et al. 2024). For the Mediterranean, the IPCC Working Group I Interactive Atlas (Gutiérrez et al. 2021) reports a median precipitation change ranging from -7.9% to -18.3% for CMIP5 and a change from -7.1% to -17.8% for CMIP6 models for different global warming levels and for pathways RCP8.5/SSP5-8.5 (relative to the pre-industrial period). For the Eastern Mediterranean and specifically Turkey, Bagçaci et al. (2021) found that the precipitation decline is 2.5% smaller and the temperature increase is up to 0.35°C higher in CMIP6 compared to CMIP5. Despite the differences reported for the latest CMIP6- relative to CMIP5-GCMs, regionally downscaled CMIP6 models are not produced at the time of preparation of this manuscript, which does not allow a direct comparison with CMIP5 RCMs (EURO-CORDEX) used here. This work showed that bias correcting the RCM of CMIP models is necessary for the quantification of regional impacts of climate change, as shown by the errors with and without bias correction of RCM outputs for annual totals and seasonal distribution of precipitation, described in Section 3.3."

Gutiérrez, J.M., R.G. Jones, G.T. Narisma, L.M. Alves, M. Amjad, I.V. Gorodetskaya, M. Grose, N.A.B. Klutse, S. Krakovska, J. Li, D. Martínez-Castro, L.O. Mearns, S.H. Mernild, T. Ngo-Duc, B. van den Hurk, and J.-H. Yoon, 2021: Atlas. In: Climate Change 2021: The Physical Science Basis. Contribution of Working Group I to the Sixth Assessment Report of the Intergovernmental Panel on Climate Change [Masson-Delmotte, V., P. Zhai, et al. (eds.)]. Cambridge University Press, Cambridge, United Kingdom and New York, NY, USA, 1927–2058. 10.1017/9781009157896.021

Bağçaci, S. Ç., Yucel, I., Duzenli, E., & Yilmaz, M. T. (2021). Intercomparison of the expected change in the temperature and the precipitation retrieved from CMIP6 and CMIP5 climate projections: A Mediterranean hot spot case, Turkey. Atmospheric Research, 256, 105576. https://doi.org/10.1016/j.atmosres.2021.105576

Wu, Y., Miao, C., Slater, L., Fan, X., Chai, Y., & Sorooshian, S. (2024). Hydrological projections under CMIP5 and CMIP6: Sources and magnitudes of uncertainty. Bulletin of the American Meteorological Society, 105(1), E59-E74. <a href="https://doi.org/10.1175/BAMS-D-23-0104.1">https://doi.org/10.1175/BAMS-D-23-0104.1</a>

Ln 82 insert word "mean"? e.g. "...highlighted a MEAN annual precipitation reduction of..."

Ln 82 We will add the word "mean" to describe the annual precipitation reduction

Ln 80-100 The introduction doesn't make clear to me what the new scientific contribution this manuscript makes.

We will modify the first half of the last paragraph in the Introduction as follows: "Previous studies analysed the performance of certain objective functions in calibrating hydrological

models under a drying climate, while other studies investigated the model performance under a non-stationary climate using a single objective function. The current study investigates how multiple, commonly used objective functions and their transformed versions affect the performance of a conceptual hydrological model under both drying and wetting conditions at different thresholds of average precipitation changes. The specific objectives are: .....".

We will also add the phase of CMIP models used in the study in Ln 97: "(iii) to bias-correct, downscale and evaluate the performance of an a large 18-member RCM ensemble from CMIP5 models for streamflow simulations".

Ln 204 will be modified as: "Precipitation and daily minimum and maximum temperature were extracted for the domain of Cyprus from 18 CMIP5-based RCMs of the EURO-CORDEX ...".

**Data and methods**

My main comment with respect to methods is there is no justification for the adoption of the 5-year calibration (and validation) window length. I understand that one wants windows short enough to have distinct wet/dry phases and I understand models were selected based on calibration and validation performance but why 5-years? However, considering the principle of 'equifinality' is 5-years sufficient for calibration? Would the results/conclusions be different for a longer window length (minimum of 10 years is typically used)?

Thank you. We will add the following in Section 2.1.2 after introducing the 5-year windows.

"Previous studies used the differential split sample test, in which periods with distinct climate conditions were comprised by discontinuous sub-periods, i.e., not consecutive hydrological years (Dakhlaoui et al. 2017), or in which, multiple consecutive sub-periods were defined by a 3-month distance from each other (Guo et al. 2020). Our study used a more balanced approach of sampling moving 5-year periods. The adoption of the 5-year calibration window length was based on a compromise between, on the one side, having long enough windows to capture climate and hydrological variability in the Mediterranean climate from year to year, and on the other side, having as many as possible sub-periods for inter-comparison. Our approach allows year-to-year dynamics to be captured in the continuous 5-year calibration and validation periods. To examine the model performance potential degradation with the 5-year calibration against an 18-year calibration, a comparison of the average NSE and KGE for all 5-year calibrations against the values of the two measures from an 18-year calibration was made. This comparison showed that the median NSE (0.82 and 0.84, respectively) and KGE (0.89 and 0.90, respectively) for the 38 watersheds were very close for the two lengths of the calibration periods. The difference in the value of the CSS was also small for the two periods (Figure R1). Based on this comparison, it can be concluded that using a longer calibration period than the 5years, will not lead to significantly better model performance also in regards to the streamflow projections with the model simulations for future periods."

**Supporting figure:**

Figure R1: The Composite Scaled Score (CSS), shown in boxplots capturing the variability of CSS among the 38 watersheds, for each of seven objective functions and for different sets of calibration and validation periods. Top: Two types of periods for calibration (long calibration – 18 years and short calibration 5 years) and validation (long validation – 17 years and short validation – 5 years), and, Bottom: Four five-year validation periods corresponding to the four classes of precipitation changes.

Ln 106 There are only two transformations not three. i.e. "1) no transformation; 2..."

This line is corrected as follows: "These functions are based on the NSE and KGE criteria, each computed with the original formula of the criteria and with two types of transformation of the streamflow values: 1) no transformation, 2) square root and 3) natural logarithm (NSE, NSEsqrt, NSElog, KGE, KGEsqrt and KGElog)."

Ln 127 Not clear how the validation were undertaken. Did they also have a 1 year warm up period?

Please refer to our response above for Ln 18.

Ln 149 method not methodology. Methodology is a study a methods (e.g. a study of different farming systems is a methodology).

Thank you for this comment. We will change "methodology" to "method".

Ln 156 I think this needs rewording as I don't know what is a "..typical annual and interannual variability in precipitation of Mediterranean climates"? South-eastern Australia and South Africa have mediterranean climates and they are among the most variable in the world.

We will modify the text as follows: "The climate of the island is characterized by the typical variability in precipitation in terms of a clear seasonality pattern, as well as, the year-to-year variability in the distribution of precipitation in the wet period found in Mediterranean climates (Hoerling et al., 2012). December and January are the wettest months; about 80% of total annual precipitation occurs between November and April."

Ln 159 I assume these are all unregulated with minimal landuse change over the experimental period? This isn't stated anywhere.

We will add the following text in Ln 159: "The 38 streamflow gauges defining the areas of the studied watersheds were selected such that any large dams and major waterworks are located downstream of the gauges. Common land use changes occurring in the studied area include agricultural land abandonment and fires over shrubland and forests. Subsequently, a conversion of the abandoned and burned area occurs, through shrub expansion and a slow (re-)growth of, predominantly, pine trees in areas where soil and rain conditions favour their development".

We will also add the following statement in Ln 186: "Streamflow observations from Kryos watershed in the southern slopes of the Troodos mountains (r9-6-2-90, see Supplemental Material) were also corrected by taking into account monthly water diversions from Arminou Dam (not included in the study area) to Kryos watershed, which started in the hydrological year 1998 – 1999. The data correction periods spans outside the calibration period."

**Results**

Figure 3 it is difficult to see change in the heat map. Can the gradient be modified to better show changes (ie introducing a third colour into the colour ramp?)

Thank you for this suggestion. We will modify the colour ramp (blue and yellow) to include three colours: blue, light yellow and brown.

Ln 366 dam storage? I think dam yield would be more appropriate? Besides changes in runoff and changed in dam yield under future climate projections are not always the same, so knowledge of the former doesn't necessarily translate to the latter.

Thank you for this comment. We will remove the mention to dam storage and modify the sentence: "To understand the implications for water resources management, such as dam storage and irrigation supply, Table 3 shows..." to "Table 3 shows...".

In addition, the following comment will be added in the Discussion section in Ln 501: "The projected increase in ET and reduction in streamflow imply reductions of equal or higher magnitude in the natural replenishment of the surface water resources of the island, including dam yield and groundwater recharge."

Figure 4 Reference evaporation is designated as ET in this manuscript, however, in this figure PET is used?

Figure 4 will be revised as below.

**Discussion**

My understanding is that this was based on CMIP5 data, how does CMIP5 data compare to CMIP6 data for this region? A brief discussion would be useful to put results of this manuscript into context of more recent CMIP6 data.

Thank you. A brief discussion will be added; please see our response from the comment above for Ln 80 and for Ln 80-100.

Ln 520 Yes this is true for mid-to-high flows there is more uncertainty in future climate inputs but for low-flows it has been found that there is more uncertainty in the hydrological models than climate inputs e.g. See Petheram et al. (2012), Teng et al. (2012),

**References**

Petheram C, Rustomji P, McVicar TR, Cai WJ, Chiew FHS, Vleeshouwer J, Van Niel TG, Li LT, Creswell RG, Donohue RJ, Teng J, and Perraud J-M (2012) Estimating the impact of projected climate change on runoff across the tropical savannas and semi-arid rangelands of northern Australia. Journal of Hydrometeorology. 13(2), 483-503, doi:10.1175/jhm-d-11-062.1; (IF 3.573; GSC: 5).

Teng J, Vaze J, Chiew F, Wang B, Perraud J-M (2012) Estimating the relative uncertainties sourced from GCMs and hydrological models in modelling climate change impact on runoff. Journal of Hydrometeorology 13(1), 122-139, doi: https://doi.org/10.1175/JHM-D-11-058.1

**We will adjust Ln 520 as follows:**

"The emphasis was placed on using multiple RCMs rather than multiple hydrological models as previous studies have shown that variability in rainfall—runoff model outputs is greater for mid- to high-flow and mean annual flow conditions when runoff projections are based on a single rainfall—runoff model combined with multiple climate models, rather than the reverse (Teng et al. 2012; Petheram et al. 2012). Other studies also showed that a significant source of uncertainty comes from the RCMs for studying climate impacts on streamflow (Teutschbein and Seibert 2012). "

---

## Author Comment (AC3)

**Response by the Authors to Reviewer 2 for EGUSPHERE-2025-2478**

We would like to thank the reviewer for the valuable recommendations to improve this manuscript and its title. The original reviewer comments are presented in black font, and our responses are in blue font.

In their manuscript, Sofokleous et al. aim to investigate the impacts of climate change on streamflow in Cyprus. To this end, the authors test various metrics for model calibration. While their general scientific aim is valid and fits well with HESS, the article's current scientific quality requires major revisions. A revision is necessary to address the article's framing, the misleading statements in the abstract, and to provide a more precise reflection of the methodology and the terminology used to explain it.

First, I suggest the title be revised. You cannot evaluate the performance of objective functions – they are means towards evaluation. It can be a comparison of different objective functions for assessing different regional climate models. And then the focus is clearly on Cyprus, not the eastern MED; otherwise, this is highly misleading.

In general, it would also be beneficial if the authors reflect on the additional value of not only assessing model performance compared to the past, but also conducting a sensitivity analysis. For example, Wagener et al. (2022) (https://wires.onlinelibrary.wiley.com/doi/full/10.1002/wcc.772) explain how response-based evaluation can be a complementary strategy. Connected to this matter is the lack of a sensitivity analysis to prepare for calibration. Without knowledge of what parameters can and should be changed, there is limited value in comparing the calibration to different metrics. The calibration also requires the authors to state which parameters are changed clearly and the corresponding value ranges. This also requires providing evidence on what parameters should be changed in the first place, i.e., a sensitivity analysis.

In response to the reviewer's suggestions, we expand our model evaluation framework to include an analysis of the parameter values obtained under different objective functions, calibration periods (average, wet, and dry), and across the different watersheds. This analysis and a new figure help clarify how sensitive the model parameterization is to the choice of objective function and hydroclimatic conditions. This additional analysis is presented in the corresponding sections below.

We also change the manuscript title, from:
*"Evaluating the Performance of Objective Functions and Regional Climate Models for Hydrologic Climate Change Impact Studies: A Case Study in the Eastern Mediterranean"*
to
*"Evaluating Objective Functions and Regional Climate Models for Hydrologic Climate Change Impact Studies: A Case Study in the Eastern Mediterranean Island of Cyprus"*

**Specific comments**

12: Compromised is the wrong word here. First, what does robustness mean here? Is it the ability to simulate with a similar skill under different circumstances? If that is the case, non-stationarity is something climate change is causing and might be a challenge, but it is not something that undermines model performance; rather, it questions whether models are fit for the right purpose.

Thank you. We will modify the sentence as follows:

"The use of hydrological models for projecting future freshwater resources is challenged by non-stationary climate conditions, as these conditions may affect whether models calibrated under historical climates are fit for use under future scenarios."

16: But why would you evaluate objective functions? To check whether your calibration is good? However, that does not test the objective function; it tests how well your model was calibrated using different objective functions.

We will change this sentence to the following: "A comparative scheme was developed to investigate how the hydrological model performs when calibrated using six different objective functions."

21: This is a misleading statement; you are simulating catchments in Cyprus, not in the MED. Similarly, your conclusion in line 30 is highly misleading as well.

We will correct the reference of "Mediterranean watersheds" in L21 to "the studied watersheds".

40: Kang Ji 2023 Reference missing

Thank you. We will correct the citation from "Kang Ji et al. (2023)" to "Ji et al. (2023)", and we will add the missing reference:

Ji, H. K., Mirzaei, M., Lai, S. H., Dehghani, A., & Dehghani, A. (2023). The robustness of conceptual rainfall-runoff modelling under climate variability–A review. Journal of Hydrology, 621, 129666. https://doi.org/10.1016/j.jhydrol.2023.129666

55: This seems like a terrible idea. Why would you restrict model simulations to such a subjective space? Unfortunately, I am not able to find the publication in the references to understand this in more detail.

We will clarify and correct L55 and move the revised sentence after the sentence in L46, as follows:

The review study of Ji et al. (2023) showed that, for credible model transferability between calibration and validation, the reported range for precipitation change towards drying conditions, by different studies, is narrower (i.e., from -10% to -30%) than the corresponding range for wetter conditions (i.e., from +10% to +80%).

56: Could you reflect on multi-objective function calibration here as well? Does this solve some of the problems, and if not, why not?

We will add the following sentence in L57:

"A composite objective function, comprised of a linear combination of different functions accounting for different aspects of flow regimes (Zhang et al. 2008), was tested against commonly used objective functions by both Fowler et al. (2018) and Munoz-Castro et al. (2023). Fowler et al. (2018) found that the multi-objective function was outperformed by simpler formulations, such as the Refined Index of Agreement (Wilmott et al., 2011), which uses the sum of absolute errors, and Kling-Gupta Efficiency (KGE; Kling et al., 2012), computed individually per year. According to these authors, these two functions also performed better than models calibrated on squared-error based measures."

Reference: Zhang, L., Potter, N., Hickel, K., Zhang, Y., & Shao, Q. (2008). Water balance modeling over variable time scales based on the Budyko framework–Model development and testing. Journal of Hydrology, 360(1-4), 117-131. https://doi.org/10.1016/j.jhydrol.2008.07.021

81: Please specify how the projections differ for the different RCPs with respect to the study ranges you cite here.

We will modify the sentence in L81 as follows:

"CMIP5 climate projections under different Representative Concentration Pathways (RCPs) for the entire Mediterranean show temperature increases of 1.1, 2.2 and 4.4 °C under RCP2.6, RCP4.5 and RCP8.5, respectively. Precipitation changes are projected to be +1%, –6.6% and –18.8% for these scenarios over the long-term period (2081–2100), relative to 1986–2005 (Gutiérrez et al. 2021). With high-resolution (0.11° ≈ 12km) RCM simulations for the Mediterranean under RCP8.5, Zittis et al. (2021a) highlighted an annual precipitation reduction of up to 10% for the first half of the 21st century and reductions of up to 20%-40% for the second half, particularly for the southern and eastern areas."

New reference:

Gutiérrez, J.M., R.G. Jones, G.T. Narisma, L.M. Alves, M. Amjad, I.V. Gorodetskaya, M. Grose, N.A.B. Klutse, S. Krakovska, J. Li, D. Martínez-Castro, L.O. Mearns, S.H. Mernild, T. Ngo-Duc, B. van den Hurk, and J.-H. Yoon, 2021: Atlas. In: Climate Change 2021: The Physical Science Basis. Contribution of Working Group I to the Sixth Assessment Report of the Intergovernmental Panel on Climate Change [Masson-Delmotte, V., P. Zhai, et al. (eds.)]. Cambridge University Press, Cambridge, United Kingdom and New York, NY, USA, 1927–2058. 10.1017/9781009157896.021

91: Performance limits in what regard?

We will remove "performance limits" and modify sentence in L91 as follows:

"Assessing the impact of climate change on fresh water resources through hydrological modelling requires understanding model performance under a range of climate-change signals and magnitudes."

96: Again, you cannot evaluate the performance of an objective function. You can evaluate the model performance of a model that has been calibrated with a specific objective function or a set of objective functions.

We will change the specific terminology and replace it accordingly in the entire manuscript. Some indicative examples follow:

In Ln96: The specific objectives are: (i) to develop a method for evaluating the performance of  a hydrological model  when calibrated with different objective functions under a changing climate.

Ln 246: "The comparison of the model performances achieved with  the six objective functions, based on the median values of the 38 watersheds (averaged for all 5-year periods) for each performance measure is shown in Table 1."

Ln 258: In the validation, the NSEsqrt-calibrated model also outperformed the other objective functions calibrated models in 14 watersheds, whereas the KGE-calibrated model parameterization outperformed the other objective functions calibrated models in three watersheds only, which is the second worst performance out of the six functions optimizations.

107: Why are you using these specific metrics and their transformations? What kind of behavior space are you covering with them? Why are you not also separating them into components that would specifically tell us something about the mass balance, peak behavior, etc.? And wouldn't this be more valuable to assess in a sensitivity analysis? This would also tell us which parameters are sensitive under which objective functions for different catchments in your assessment.

Thank you for your comments.

Concerning the selection of the specific metrics, we will add the following justification in Ln107:

"NSE and KGE were selected as the main objective functions due to their widespread use in hydrological modelling in both original and modified forms (Fowler et al. 2018; Guo et al. 2020). The use of the original, square-root, and logarithmic transformations ensures representation of high-, medium-, and low-flow conditions (Moriasi et al. 2007; Seiller et al. 2017). Percent bias (PBIAS) was also included to provide an independent measure of total volume error (Moriasi et al. 2007; Coron et al. 2012)."

In order to highlight the importance of conducting a sensitivity analysis of model components that might affect the results of the impact study we will add the following sentence in introduction in L55:

"A sensitivity analysis of the model response to multiple inputs provides useful insight into model behaviour under changing conditions, as recommended by Wagener et al. (2022)."

We will also add a model parameter analysis that shows the parameter value ranges under different objective functions, different calibration periods and for different watersheds. The results of this analysis will be added to the end of Section 3.1 "Selection of objective function under changing climate conditions", as follows:

"The optimized model parameters derived using different objective functions and calibration periods were plotted against the average precipitation of the respective calibration periods (Figure A1). The results show that, for a given watershed, parameter values generally vary more between different objective functions than between different calibration periods. This finding suggests that understanding how different objective functions lead to different model outputs, in relation to the input model parameters and interannual variations in precipitation forcing, is important."

[Figure]

Figure A1: Values of the four parameters of the GR4J model optimized with six objective functions for the 15 five-year calibration periods for eight watersheds. The mean precipitation of each five-year calibration period is shown on the horizontal axis.

New reference:

Wagener, T., Reinecke, R., & Pianosi, F. (2022). On the evaluation of climate change impact models. Wiley Interdisciplinary Reviews: Climate Change, 13(3), e772. https://doi.org/10.1002/wcc.772

135: How sensitive is the calibration to picking this specific value? How does this assumption impact the results?

We will add the following explanation in L136:

"The minimum temperature increase used to select calibration and validation runs was 0.7°C. This value did not exclude any five-year period from being tested for calibration, because every five-year window within the calibration period (1980–1997) could be matched with at least one five-year window in the validation period (1998–2015) that was warmer by 0.7°C or more."